Continental-scale suppression of an invasive pest by a host-specific parasitoid underlines both environmental and economic benefits of arthropod biological control

Wyckhuys Kris A.G. kagwyckhuys@gmail.com 1 2 3 4 19
Wongtiem Prapit 5
Rauf Aunu 6
Thancharoen Anchana 7
Heimpel George E. 8
Le Nhung T.T. 9
Fanani Muhammad Zainal 6
Gurr Geoff M. 1 10
Lundgren Jonathan G. 11
Burra Dharani D. 12
Palao Leo K. 12
Hyman Glenn 13
Graziosi Ignazio 14 15
Le Vi X. 9
Cock Matthew J.W. 16
Tscharntke Teja 17
Wratten Steve D. 1 18
Nguyen Liem V. 9
You Minsheng 1
Lu Yanhui 19
Ketelaar Johannes W. 20
Goergen Georg 21
Neuenschwander Peter 21
1 Institute of Applied Ecology, Fujian Agriculture & Forestry University , Fuzhou , Fujian , China
2 School of Biological Sciences, University of Queensland , Brisbane , Australia
3 Institute of Insect Sciences, Zhejiang University , Hangzhou , China
4 CGIAR Program on Roots, Tubers and Banana, International Center for Tropical Agriculture , Hanoi , Vietnam
5 Rayong Field Crops Research Center, Thai Department of Agriculture , Rayong , Thailand
6 Bogor Agricultural University , Bogor , Indonesia
7 Kasetsart University , Bangkok , Thailand
8 University of Minnesota , Minneapolis , United States of America
9 Plant Protection Research Institute , Hanoi , Vietnam
10 Charles Sturt University , Orange , Australia
11 Ecdysis Foundation , Estelline , United States of America
12 International Center for Tropical Agriculture CIAT , Hanoi , Vietnam
13 International Center for Tropical Agriculture CIAT , Cali , Colombia
14 University of Kentucky , Lexington , United States of America
15 World Agroforestry Center ICRAF , Nairobi , Kenya
16 CABI , Wallingford , United Kingdom
17 University of Goettingen , Goettingen , Germany
18 Lincoln University , Christchurch , New Zealand
19 Institute of Plant Protection, China Academy of Agricultural Sciences , Beijing , China
20 Food and Agriculture Organization , Bangkok , Thailand
21 International Institute of Tropical Agriculture , Cotonou , Benin
Montoya Jose
Electronic publication date: 2018 Oct 19
Publication date: 2018
Volume: 6
Electronic Location ID: e5796
Received 2018 Jun 13; Accepted 2018 Sep 20
Copyright: ©2018 Wyckhuys et al.
Copyright year: 2018
Copyright holder: Wyckhuys et al.
License: This is an open access article distributed under the terms of the Creative Commons Attribution License, which permits unrestricted use, distribution, reproduction and adaptation in any medium and for any purpose provided that it is properly attributed. For attribution, the original author(s), title, publication source (PeerJ) and either DOI or URL of the article must be cited.
License URL: https://creativecommons.org/licenses/by/4.0/

Keywords: Ecosystem services, Ecological intensification, Insect biological control, Tropical agro-ecosystems, Sustainable agriculture, Invasion biology, Ecological safety, Insect parasitism

Funding: CIAT-executed programme CIAT-EGC-60-1000004285 CGIAR-wide Research Program Tubers and Banana (CRP-RTB) Kris AG Wyckhuys This initiative was conducted as part of an EC-funded, IFAD-managed, CIAT-executed programme (CIAT-EGC-60-1000004285), while additional funding was provided through the CGIAR-wide Research Program on Roots, Tubers and Banana (CRP-RTB). Activities in Thailand and Indonesia during 2017 were finalized using own resources of Kris AG Wyckhuys. There was no additional external funding received for this study. The funders had no role in study design, data collection and analysis, decision to publish, or preparation of the manuscript.

==============================
Biological control, a globally-important ecosystem service, can provide long-term and broad-scale suppression of invasive pests, weeds and pathogens in natural, urban and agricultural environments. Following (few) historic cases that led to sizeable environmental up-sets, the discipline of arthropod biological control has—over the past decades—evolved and matured. Now, by deliberately taking into account the ecological risks associated with the planned introduction of insect natural enemies, immense environmental and societal benefits can be gained. In this study, we document and analyze a successful case of biological control against the cassava mealybug, Phenacoccus manihoti (Hemiptera: Pseudococcidae) which invaded Southeast Asia in 2008, where it caused substantial crop losses and triggered two- to three-fold surges in agricultural commodity prices. In 2009, the host-specific parasitoid Anagyrus lopezi (Hymenoptera: Encyrtidae) was released in Thailand and subsequently introduced into neighboring Asian countries. Drawing upon continental-scale insect surveys, multi-year population studies and (field-level) experimental assays, we show how A. lopezi attained intermediate to high parasitism rates across diverse agro-ecological contexts. Driving mealybug populations below non-damaging levels over a broad geographical area, A. lopezi allowed yield recoveries up to 10.0 t/ha and provided biological control services worth several hundred dollars per ha (at local farm-gate prices) in Asia’s four-million ha cassava crop. Our work provides lessons to invasion science and crop protection worldwide. Furthermore, it accentuates the importance of scientifically-guided biological control for insect pest management, and highlights its potentially large socio-economic benefits to agricultural sustainability in the face of a debilitating invasive pest. In times of unrelenting insect invasions, surging pesticide use and accelerating biodiversity loss across the globe, this study demonstrates how biological control—as a pure public good endeavor—constitutes a powerful, cost-effective and environmentally-responsible solution for invasive species mitigation.

Introduction

Biological control is a globally-important ecosystem service, and plays a pivotal role in the functioning and broader resilience of agricultural and natural ecosystems alike (Costanza et al., 1997). For US agriculture alone, insect-mediated biological control is conservatively valued at $4.5–17 billion per year, and a diverse community of natural enemies helps alleviate pressures from herbivores and other crop antagonists (Losey & Vaughan, 2006). However, rapid depletion of animal populations and progressive ecosystem simplification compromise the strength and stability of this ecosystem service (Oliver et al., 2015; Hallmann et al., 2017). In tropical terrestrial ecosystems with inherently high levels of biodiversity, these trends might be even more pronounced yet routinely remain un-documented due to e.g., an overall lack of sufficient funds, scientific interest and well-trained ecologists (Melo et al., 2013; Barnes et al., 2014).

Across the globe, arthropod pests reduce agricultural productivity by 10–16% and constitute key impediments to food security and (indirectly) poverty alleviation (Oerke, 2006; Bebber, Ramotowski & Gurr, 2013). Though native pests continue to pose major problems for the world’s agriculture, non-native species are of increasing significance as a result of trade globalization and human movement (Bradshaw et al., 2016; Paini et al., 2016). Importation biological control (IBC; also known as ‘classical biological control’), or the judicious selection and subsequent introduction of a specialized natural enemy from the pest’s region of origin, has been repeatedly shown to effectively reduce invasive pest populations to less damaging levels (Van Driesche, Hoddle & Center, 2008; Heimpel & Mills, 2017). Particularly in the developing-world tropics, IBC can be a “silver bullet” option for destructive agricultural pests, being largely self-sustaining and requiring little or no stakeholder intervention (Andrews, Bentley & Cave, 1992). Since the late 1800s, more than 2,000 natural enemy species have been released against approximately 400 invasive pests worldwide, occasionally resulting in complete pest control but regularly causing limited impact on target pest populations (Van Lenteren et al., 2006; Cock et al., 2016b). Though economic impacts are not routinely assessed for IBC, levels of pest suppression and ensuing benefit:cost ratios can be exceptionally favorable (5:1 to >1,000:1) (Heimpel & Mills, 2017; Gutierrez, Caltagirone & Meikle, 1999; Naranjo, Ellsworth & Frisvold, 2015). Yet, IBC is marred with remarkably low rates of success (Greathead & Greathead, 1992; Cock et al., 2016a), and biological control as a whole is habitually undervalued or even avoided while it should be promoted (Daily et al., 2009). Furthermore, over the past three decades, IBC initiatives have been met with stringent regulations and a heightened emphasis on potential ecological risks or unintended side-effects (Heimpel & Cock, 2018). The latter was triggered by a provocative yet necessary account by Howarth (1983) and Howarth (1991), built around misguided biological control releases that were conducted decades earlier, and in which the long-established paradigm of IBC as ‘ecologically-safe’ practice was challenged.

One well-recognized IBC program is the Africa-wide initiative targeting the invasive cassava mealybug, Phenacoccus manihoti (Hemiptera: Pseudococcidae), which led to a 50% yield recovery resulting in long-term economic benefits up to US $20.2 billion as well as the likely avoidance of widespread famine without negative side effects (Neuenschwander et al., 1989; Herren & Neuenschwander, 1991; Zeddies et al., 2001). Key to the success of this program was the carefully-selected host-specific and environmentally-adaptable parasitoid Anagyrus lopezi (Hymenoptera: Encyrtidae), recovered in 1981 after foreign exploration from South America, and introduced into Nigeria soon thereafter. As A. lopezi is considered to be a specialist internal feeder on P. manihoti, no detrimental ecological impacts resulted from its continent-wide release (Neuenschwander, 2001). Following its devastating passage through Africa’s cassava belt in the 1970s and 80s, P. manihoti was inadvertently introduced into Thailand in 2008, spread through mainland Southeast Asia, and made its appearance in Indonesia by 2010 (Graziosi et al., 2016). As cassava is grown on >4 million ha by an estimated 8 million farming families throughout tropical Asia, this pest had potential to cause massive socio-economic impacts. As part of an internationally-coordinated management campaign for P. manihoti, A. lopezi was promptly sourced from Benin, West Africa and 500 adult parasitoids were introduced into Thailand in 2009 (Winotai et al., 2010). Parasitoids were subsequently mass-reared by multiple Thai institutions, released across the country during 2010–2012 (some by airplane) and introduced into neighboring Laos, Cambodia (in 2011), Vietnam (in 2013) and Indonesia (in 2014) (Wyckhuys, Rauf & Ketelaar, 2015).

In this study, we characterized the degree to which A. lopezi has established in the highly-heterogeneous cassava cropping environments of Southeast Asia. Field research was carried out over the course of 2014-2017 by various country teams, each pursuing different objectives as outlined below. We employed seasonal population surveys that extended from Myanmar’s Ayeyawaddy River delta to the uplands of Timor in eastern Indonesia, to quantify magnitude and spatial extent of parasitoid-induced P. manihoti population suppression (section i, ii). Furthermore, we employed well-established manipulative protocols to assess the effectiveness of A. lopezi and subsequent yield benefits of biological control (DeBach & Huffaker, 1971; Van Lenteren, 1980; Luck, Shepard & Kenmore, 1988) (section iii). Finally, we conducted an analysis of production statistics and cassava prices in one of Asia’s main cassava-growing countries (Thailand) over a time period spanning the 2008 P. manihoti invasion, the 2009 parasitoid introduction into Thailand and the subsequent (natural, and human-aided) continent-wide distribution of A. lopezi (section iv).

Our work uses original datasets to present a rare, continental-scale and multi-year assessment of IBC-mediated insect pest suppression, and the cascading trophic and socioeconomic effects on cassava yield loss reduction and commodity prices. We combine observational and manipulative studies to exemplify the benefits of A. lopezi as a biological control agent, and lay the basis for further econometric investigations. This study illustrates the value of an insect-driven ecosystem service to support agricultural sustainability, in the face of a potentially devastating invasive pest.

Materials & Methods

Multi-country pest & natural enemy survey

From early 2014 until late 2017, insect surveys were carried out in 601 cassava fields in Myanmar, Thailand, Lao PDR, Cambodia, Vietnam, southern China and Indonesia. Survey protocols are described in detail in Graziosi et al. (2016). In brief, we selected older fields (i.e., 8–10 months of age) in the main cassava-growing areas of each country, with individual sites located at least 1 km apart. Five linear transects were randomly chosen per site, with ten plants (routinely spaced at 0.8–1.2 m) sampled in each transect. By doing so, a total of 50 plants per field were assessed for P. manihoti infestation and per-plant mealybug abundance. In-field identification of mealybugs was based on morphological characteristics such as coloration and presence or length of abdominal waxy filaments, while samples were also taken to the laboratory for identification by specialist taxonomists. Following transect walks we calculated average P. manihoti abundance (i.e., number of individuals per infested tip) and field-level incidence (i.e., proportion of P. manihoti-infested tips per field).

To assess local A. lopezi establishment and parasitism rates, we conducted dry-season sampling from 2014 to 2017 at sub-sets of mealybug-invaded sites in Thailand (n = 20), Cambodia (n = 10, 15), southern Vietnam (n = 20, 20, 6) and Indonesia (n = 10, 9, 21) (total n = 131). Sampling consisted of collecting 20 mealybug-infested tips from local fields and transferring them to a laboratory to monitor subsequent parasitoid emergence (Neuenschwander et al., 1989). Surveys were carried out during January-May 2014 (dry season), October–November 2014 (late rainy season), January–March 2015 (dry season) in mainland Southeast Asia, and during October–November 2014 and 2017 (dry season) in insular Indonesia. Locations were recorded using a handheld GPS unit (Garmin Ltd, Olathe, KS). In-field identification of mealybugs was based on morphological characters, while samples were also transferred to the laboratory for further taxonomic identification. Voucher specimens of P. manihoti were equally deposited at the Thai Department of Agriculture (Bangkok, Thailand), Bogor Agricultural University (Bogor, Indonesia) and Plant Protection Research Institute (Hanoi, Vietnam).

To assess local A. lopezi establishment and parasitism rates, mealybug-infected tips were collected in the field and transferred to a laboratory. Upon arrival in the laboratory, each tip was carefully examined, predators were removed and the total number of P. manihoti was determined. Tips were then placed singly into transparent polyvinyl chloride (PVC) containers, closed with fine cotton fabric mesh. Over the course of three weeks, containers were inspected on a daily basis for emergence of parasitoids and A. lopezi parasitism levels (per tip and field) were computed. Next, for fields where presence of A. lopezi was reported, we carried out a regression analysis to relate field-level mealybug abundance with parasitism rate. Mealybug infestation levels and parasitism rates were log-transformed to meet assumptions of normality and homoscedasticity, and all statistical analyses were conducted using SPSS.

Multi-year mealybug and parasitoid population assessment in Vietnam

From July 2013 until July 2015, we conducted population surveys in Tay Ninh province, Vietnam; an area with near-continuous, all-year cassava cultivation (see also Le et al., 2018). The cassava mealybug is assumed to have arrived in southern Vietnam during 2011–2012, and A. lopezi was first detected from Tay Ninh province in early 2013. Eight newly-planted cassava fields were selected of uniform age, crop variety, developmental stage and management. Every two months, insect surveys were done within these fields to characterize P. manihoti incidence, infestation pressure and A. lopezi parasitism rate. In each field, a total of five linear 10–15 m transects were screened (plants routinely spaced at 0.8–1.2 m) and, 50 plants were carefully inspected for P. manihoti. Phenacoccus manihoti infestation was recorded as field-level abundance (number of individuals per infected tip) and field-level incidence (proportion of mealybug-affected tips) at each sampling date and location. To assess A. lopezi parasitism rates, 20 mealybug-infested tips were randomly collected from each field by breaking off the top parts of individual plants, and transferred to the laboratory. Parasitism rates were estimated from these samples as described above, and parasitism levels were computed for each individual field and sampling date. We used analysis of variance (PROC MIXED, SAS version 9.1; SAS Institute, Cary, NC, USA) with field as the random factor, and tested the effect of cassava age, sampling date and year for P. manihoti incidence, abundance and A. lopezi parasitism. Means were compared with least squares means approach. Mealybug abundance data were log-transformed while incidence, parasitism and hyperparasitism data were arcsine-transformed to meet normality.

The intrinsic rate of mealybug population increase, r, over 2 months was calculated over subsequent sampling events as ln(mt+1/ mt) where m = the per-tip mealybug density. This growth rate was regressed against the mealybug parasitism rate as a means of evaluating the role of the parasitoids in suppressing mealybug population growth rates and also of estimating the parasitism level that correlates with a decrease in mealybug population growth rate (for other uses of this approach, see Lin & Ives, 2003; Plecas et al., 2014). The statistical significance of the relationship between parasitism rate and mealybug population growth was assessed using a generalized linear model incorporating normal error distribution with r as the response variable and parasitism level and field identity as independent variables.

Exclusion cage assays

In August 2014, a field study was initiated at the Rayong Field Crops Research Center (RYFCRC) in Rayong, Thailand (see Thancharoen et al., 2018). To assess the relative contribution of natural enemies such as A. lopezi to pest control, we employed exclusion assays (Snyder & Wise, 2001; Costamagna, Landis & DiFonzo, 2007). More specifically, to determine separate and joint effects of P. manihoti and A. lopezi on cassava crop yield, four different treatments were established using two common cassava clones: Kasetsart 50 (KU50) and Rayong 72 (R72). Treatments consisted of the following: (1) ‘full cage’ assays, in which a plant was entirely covered by a mesh screen cage to exclude all natural enemies; (2) ‘sham’ cage assays, in which a plant was covered by a screen cage to provide a microhabitat similar to that of the ‘full cage’, but left open at the sides to allow natural enemy access; (3) ‘no cage’ assays, in which a plant was kept without a cage as a ‘real-world’ benchmark. For the ‘no cage’ assay, two different mealybug infestation modes were adopted: (1) one single infestation with 10 mealybug adults per plot at the onset of the experiment; (2) a monthly infestation with 10 P. manihoti adults per plot, to ensure a sustained mealybug population over the entire course of the experiment. Each treatment (i.e., a single cage containing four replicated plants) was established with four replicates, for two different cassava clones. The experimental field was established using locally-sourced stem cuttings of KU50 or R72, planted at 1-m distances within plots.

Once plants had reached 4.5 months of age, 2 × 2 × 2 m polyvinylchloride (PVC) frame cages were deployed, with four plants contained within each cage. Cages were covered with fine nylon mesh screen to prevent entry by insects, including A. lopezi parasitoids. In January 2015, 10 adult female P. manihoti were gently brushed onto plants within each treatment. Mealybug adults were obtained from a laboratory colony at RYFCRC that had been started in early 2014, in which P. manihoti were maintained on potted cassava plants within a screen-house that were regularly supplemented with field-collected individuals. Visual observations were carried out within the cages on a monthly basis and P. manihoti abundance was recorded on each plant. On September 7, 2015, once the crop had reached 12 months of age, cages were removed and plants within the different experimental treatments were harvested manually. At harvest, fresh root yield (FRY) was determined for each plant to determine treatment effects (Karlström et al., 2016).

Mealybug population build-up under each experimental treatment was calculated, by converting the average number of mealybugs per plant (averaged across plants within a given treatment, as to avoid pseudo-replication) on a given sampling date to cumulative mealybug-days (CMD) (Ragsdale et al., 2007): ∑n=1∞=xi−1+xi2×ti−ti−1

where n is the total number of days over which sampling took place, xi is the number of mealybugs counted on day i and ti is the number of days since the initiation of sampling on day i.

Mealybug population build-up under each treatment was computed, and average CMD measures were compared between the respective treatments using a mixed modeling approach with plot as the random factor and time as the repeated measure. A mixed modeling approach was used to contrast fresh root yield (FRY), using treatment and variety as fixed factors. Plant survival rates were compared between treatments, using a Chi-square analysis. Where necessary, data were transformed to meet assumptions of normality and homoscedasticity, and all statistical analyses were conducted using SPSS. In cases where data could not be transformed to normality, non-parametric tests (e.g., Kruskal–Wallis) were used.

Country-wide yield changes

Crop production statistics were obtained through the Office of Agricultural Economics, Ministry of Agriculture & Cooperatives (Bangkok, Thailand). Yield measures were computed for 2006–2016, for a total of 51 cassava-growing provinces within Thailand, and annual weighted means were compared between successive years. Province-level yields were assigned different weights, based upon the local extent of cassava cultivation (in terms of harvested area) during a given year. Cassava crop yield can be impacted by agro-climatic conditions (e.g., temperature-related variables) and by attack of pests such as P. manihoti. To assess the impact of sustained A. lopezi releases from the 2011 cropping season onward, mean values of yields across all the cassava-growing provinces were regressed with explanatory variables which included rainfall, minimum and maximum temperature (obtained from Thai Meteorological Department, Bangkok, Thailand). Additionally, time (year for which yield observations and agro-climatic data were obtained) was also added as an explanatory variable to control for variation in the variables across years. In addition, a categorical variable representing the introduction of A. lopezi (dummy-coded as 1 for ‘presence’ for the 2011 and 2012 growing seasons, and dummy-coded as 0 for ‘absence’ for growing seasons 2008, 2009 and 2010) was also added as an additional explanatory variable in the regression model. Before proceeding for further analysis, the distribution of the response variable (i.e., yield) was tested and was identified to be normal (Shapiro test, p < 0.05). A step-wise regression approach (forward and backward) using a linear modeling approach was used to identify the model that best explains variation in yield. The model with the lowest Akaike information criterion (AIC) was selected. In the next step, the model with the lowest AIC score was compared with models containing interaction terms between time and other explanatory variables (i.e., temperature minimum, rainfall and A. lopezi introduction) both separately and simultaneously (see Table S1). Model with the least AIC score, and the highest adjusted R2 value was selected and is described in the Results section. Regression analysis was performed in R (v 3.4.1) statistical computing environment. Additionally, R package “gvlma” was used to assess if the assumptions of regression were met by the selected model. Additional diagnostics of the selected model, such as determination of variance inflation factor (VIF) for detection of multicollinearity, the Non-constant Variance Score Test (i.e., test for heteroscedasticity of residuals over fitted values) was performed using R package “MASS” and “car”, respectively.

Results

Multi-country pest & natural enemy survey

During continental-scale insect surveys from 2014 until 2017 (i.e., 5–8 years following the initial A. lopezi introduction), the mealybug complex on cassava largely comprised four non-native species: (1) P. manihoti; (2) the papaya mealybug Paracoccus marginatus Williams & Granara de Willink; (3) Pseudococcus jackbeardsleyi Gimpel & Miller; and (4) the striped mealybug Ferrisia virgata Cockerell. Phenacoccus manihoti was the most abundant and widespread mealybug species, and was reported from 37.0% (n = 549) and 100% fields (n = 52) in mainland Southeast Asia and Indonesia, respectively. Among sites, P. manihoti reached field-level incidence of 7.6 ± 15.9% (mean ± SD; i.e., proportion mealybug-affected tips) and abundance of 14.4 ± 31.0 insects per infested tip in mealybug-affected fields (or 5.2 ± 19.8 insects per tip across all fields) in mainland Southeast Asia, and incidence rates of 52.7 ± 30.9% and 42.5 ± 67.7 individuals per tip in Indonesia. Field-level incidence and population abundance were highly variable among settings and countries, reaching respective maxima of 100%, and 412.0 individuals per tip (Fig. 1).

Figure 1 Map of Southeast Asia, depicting P. manihoti spatial distribution, infestation pressure and A. lopezi parasitism rates.

Doughnut charts in the left and right margins represent field-level incidence (i.e., red portion—on a blue background—reflecting the proportion of P. manihoti affected tips, ranging from 0 to 1 for full circumference), and are complemented with bar charts indicative of plant-level P. manihoti abundance (i.e., average number of individuals per tip). The number inside each doughnut reflects the number of fields sampled per locale. Doughnut charts in the lower panel indicate average A. lopezi parasitism rate at six selected locales (depicted by the dark green section—on a light green background—reflecting proportion parasitism ranging from 0 to 1 for full circumference), with varying numbers of fields sampled per locale. The distribution map is created as overlay on a 2005 cassava cropping area (You et al., 2017). Photograph: Anagyrus lopezi (credit G. Goergen, IITA).

When examining P. manihoti parasitism rates from a select set of sites, A. lopezi was present in 96.9% of mealybug-affected fields (n = 97) in mainland Southeast Asia, yet were only found in 27.5% sites (n = 40) across Indonesia. Among sites, highly variable parasitism rates were evident with dry-season rates of 16.3 ± 3.4% in coastal Vietnam, versus 52.9 ± 4.3% in intensified systems of Tay Ninh (also in Vietnam). In Indonesia, A. lopezi was found in 22.0% fields in Lombok (n = 9) and was absent from prime growing areas in Nusa Tenggara Timur (NTT). In sites where A. lopezi had successfully established, dry-season parasitism ranged from 0% to 97.4%, averaging 30.0 ± 24.0% (n = 110) (Fig. S1). In fields where A. lopezi had effectively established, mealybug pest pressure exhibited a negative regression with parasitism rate (F1,98 = 13.162, p < 0.001; R2 = 0.118).

Multi-year mealybug and parasitoid population assessment in Vietnam

Over the course of three years, we monitored P. manihoti abundance, field-level incidence and associated A. lopezi parasitism rates in Tay Ninh, southern Vietnam. Field-level incidence of P. manihoti ranged from 0% to 82%, averaging 24.8 ± 17.7% (mean ± SD) plants infested over two consecutive crop cycles. Mealybug incidence was significantly higher on older crops F7,57 = 9.9; p < 0.0001 ), and rapidly increased during the dry season. Similarly, mealybug abundance (average 5.6 ± 5.0 individuals per tip) was higher during the dry season (F1,63 = 9.10; P = 0.0037), and in crops older than six months, when compared to younger crops (F7,57 = 2,694.06; P < 0.0001). From mid-2014 onward, mealybug populations remained at field-level abundance below 10–15 individuals per infested tip (Fig. 2). Furthermore, A. lopezi attained mean parasitism rates of 42.3 ±  21.7%, with maxima of 76.7 ± 28.9% during the early rainy season (Fig. 2). Overall, parasitism gradually increased over the dry season, up until crops were 4–6 months old.

Figure 2 Bi-monthly mealybug population fluctuations in southern Vietnam, over a 2-year time period.

Phenacoccus manihoti dynamics are represented following the first record of A. lopezi presence in southern Vietnam, depicting field-level P. manihoti abundance (n = 8) as contrasted with respective A. lopezi parasitism rates, from July 2013 until July 2015.

Mealybug growth rates were significantly and negatively correlated with parasitism levels across the 8 sites studied (GLM w/ Normal error distribution and corrected for field: χ12=125.4; P = 0.0017; the field term was not significant; χ72=0.2; P = 1) (Fig. 3). The x-intercept of each per-field regression represents the parasitism level above which mealybug growth rates are negative and this value ranged between 0.38 and 0.69 for the 8 sites (average = 0.47  ± 0.09) (Fig. 3). Whilst A. lopezi was the sole primary parasitoid at this location, three hyperparasitoid species attacked it locally it at 2.79 ± 5.38% levels (as % of parasitized hosts).

Figure 3 Effect of cassava mealybug parasitism rate on intrinsic rate of mealybug increase over consecutive 2-month periods in Tay Ninh, Vietnam.

Each dot represents a 2-month period in one of eight field plots. Thin lines are linear regressions per each of the eight sites monitored for illustrative purposes although the analysis was done on the entire data set. The thick black line shows the fit of the entire data set. The red dotted line shows r = 0; values above this on the y axis indicate positive growth of mealybug populations and below it indicate negative population growth. Parasitism level above which P. manihoti growth rates are negative ranged between 0.38 and 0.69 for the eight sites. See text for statistical details.

Exclusion cage assays

Over the entire assay, P. manihoti populations under ‘full cage’ attained 48,318 ± 51,425 (n = 4; mean ± SD) and 7,256 ± 8,581 cumulative mealybug days (CMD) in ‘sham cage’ for one popular variety (i.e., R72) (Thancharoen et al., 2018; Fig. 4). For a second variety, KU50, P. manihoti attained 28,125 ± 32,456 CMD in a ‘full cage’ treatment, and 1,782 ± 1,073 CMD in ‘sham cage’. This compared to CMD measures in a ‘no cage’ control of 1,378  ± 1,039 and 342 ± 252, for R72 and KU50 respectively. CMD measures were significantly affected by treatment (F3,189 = 240.752, p < 0.001) and time (F6,189 = 113.347, p < 0.001), and the interaction term time x treatment (F18,189 = 2.012, p = 0.011). Also, total CMD measures at the end of the trial significantly differed between treatments for both R72 and KU 50 (F3,12 = 6.767, p = 0.006; F3,12 = 11.152, p = 0.001, respectively).

Figure 4 Mealybug abundance and subsequent yield parameters for two cassava varieties under an exclusion cage assay at Rayong, Thailand.

Six weeks after inoculation, mealybug abundance (n = 16; mean ± SE) is compared between treatments for two common varieties (R72, KU50), and is significantly higher under ‘full cage’ conditions (i.e., exclusion of natural enemies, incl. A. lopezi), as compared to ‘sham cage’ and un-caged controls (ANOVA, F2,45 = 50.289, P < 0.001 for R72; F2,45 = 9.807, P < 0.001 for KU50). For each treatment, fresh root yield is determined at time of harvest, on a 12-month old crop.

Cassava yield parameters varied considerably under the four experimental treatments, and for both crop varieties (see Thancharoen et al., 2018). For Rayong 72, plant survival attained 37.5% under ‘full cage’ as compared to 75% and 87.5% under ‘no cage’ or ‘sham cage’ conditions, respectively (Chi square, χ2 = 10.473, p = 0.015). Fresh root yield (FRY) was significantly affected by treatment (F3,27 = 4.104, p = 0.016) and variety (F1,27 = 4.364, p = 0.046). For R72 and KU50, FRY under ‘full cage’ was 74.6% or 71.2% lower than under ‘sham cage’ (Kruskal–Wallis, χ2 = 8.344, p = 0.039; χ2 = 19.134, p < 0.001, respectively), and respective yield reductions for both varieties were 77.2% and 67.8% compared to ‘no cage’ treatments.

Country-wide yield changes

During the 2009 dry season, P. manihoti attained its peak population in Thailand, with field-level incidence near 100% and abundance rates of hundreds of P. manihoti per plant on at least 230,000 ha (Rojanaridpiched et al., 2013). Over the subsequent 2009-10 cropping season, province-level crop yields dropped by 12.59 ± 9.78% nationwide (weighted mean: −18.2%) (Fig. 5). Furthermore, country-wide aggregate yields declined from 22.67 t/ha to 18.57 t/ha, and total production dropped by 26.86% to 22,005,740 tonnes of fresh root. Following the lowered crop output, prices for Thai cassava starch increased 2.38-fold at domestic prices in Thailand, and 2.62-fold at export prices (US$ FOB, Free On Board) (Fig. S2). However, in the following growing seasons (i.e., 2011 and 2012), there was a marked improvement in yield across all regions, as compared to previous years (Fig. 5). It was also from mid-2010 onward that country-wide mass releases of A. lopezi were carried out. To test if the A. lopezi introduction was responsible for this improvement in yield, and to differentiate P. manihoti-induced yield drops from agro-climatic climatic impacts and changes across years, regression analyses were carried out with yield as response variable and agro-climatic parameters, year (“time”), and A . lopezi presence as explanatory variables. Multiple regression analysis revealed that a model with interaction terms between time and all explanatory variables, i.e., time of introduction of A. lopezi and rainfall had the lowest AIC score and the higest explanatory power (adjusted R2) (Table S1). The model showed a significantly positive effect (F7,183 = 8.641) of the interaction term Time ×Presence (i.e., ‘presence’ of A. lopezi and time , p < 0.01) on observed yields. The best model indicated that the introduction of A. lopezi (and not agro-climatic variables or changes in those variables over time) significantly increased yields across all cassava-growing regions during the 2011 and 2012 cropping seasons.

Figure 5 Annual percent shifts in crop yield (t/ha) for 51 cassava-growing provinces in Thailand, reflective of the mealybug invasion and ensuing biological control.

Shifts (A–D) cover the country-wide spread of P. manihoti from late 2008 until 2011, the first small-scale release of A. lopezi (Nov. 2009) and subsequent nation-wide distribution of the parasitoid from June 2010 onward. Province-level yield shifts depict the percent change of crop yield in one given year, as compared to the previous year.

Discussion

In 2008, the invasive mealybug P. manihoti accidentally arrived in Thailand. Through its extensive spatial spread, rapid population build-up and unrestricted feeding on plants (this leading to stunting and plant death), P. manihoti caused considerable yield declines and a 27% drop in the nation’s cassava production. This study shows how the neotropical parasitoid, A. lopezi, released throughout Thailand for mealybug control in 2010, had effectively established in 97% mealybug-affected fields in mainland Southeast Asia by 2014, and colonized 27% sites across insular Indonesia by late 2017. Attaining average dry-season parasitism rates of 30% across sites, A. lopezi populations readily oscillate with those of its mealybug host and suppress P. manihoti to incidence levels of 7% and background infestation pressure of a mere 14 individuals per (infested) tip. Experimental assays using two widely-grown cassava varieties reveal how biological control secures approximate yield gains of 5.3–10.0 t/ha. Our work demonstrates how A. lopezi downgrades the invasive P. manihoti to non-economic levels at a continental scale, without any known detrimental side-effects. Offering a quantitative assessment of IBC’s contribution to (the recovery of) primary productivity in Asia’s cassava crop, our work illuminates the broader societal value of biological control in a geographical region where there is heavy and increasing use of pesticides (Schreinemachers et al., 2015).

Aside from featuring as a ‘beacon of hope’ in Asia’s pesticide-tainted farming systems, our work heralds a new era for the discipline of insect biological control. Since the late 1800s, biological control has permitted the complete or partial suppression of 226 debilitating insect pests globally, it has formed the crux of founding ecological theories (e.g. Hairston, Smith & Slobodkin, 1960), and was widely deemed to be a safe, dependable and preferred means for (invasive) pest control for most of its history. Following the release of Rachel Carson’s 1962 Silent Spring, biological control was met with unrestrained enthusiasm and a firm belief in its potential as a sound alternative to pesticide-centered practices. Yet, as concerns over its ecological risks rose following the Howarth (1983) and Howarth (1991) denunciation of few historic cases of malpractice, regulatory hurdles surfaced, public funding lowered and the practice of insect biological control went through trying yet necessary reform (Strong & Pemberton, 2000; Hoddle, 2004; Messing & Brodeur, 2018). Over the past three decades, IBC implementation has centered on ecological safety and increasingly strives to balance environmental benefits and risks (Heimpel & Cock, 2018). Though weed biological control indeed has a 99% safety record (Suckling & Sforza, 2014), scientists are conscious that ecological risk will never be zero and that certain risk factors are difficult to anticipate and predict (Crooks & Soulé, 1999; Sexton et al., 2017). In the meantime, it’s well-recognized that invasive pests tend to present far higher threats to native biota than judiciously-selected natural enemies with a narrow dietary breadth (Culliney, 2005). Though the 1980s Africa campaign against P. manihoti was implemented during times when the primary focus of insect biological control was on benefits (but see Neuenschwander, 2001), risks were considered to be minimal and did not delay implementation. The fact that A. lopezi was both effective and highly host-specific vindicated this. As a result, the implementation of IBC in Southeast Asia more than 30 years later was greatly facilitated by recognizing that (i) IBC had been effective across Africa’s cassava-belt, and (ii) widespread benefits were amplified because of an absence of negative environmental side-effects on non-target species.

In light of the above, A. lopezi attained consistently high parasitism rates across most of the P. manihoti range of climatic suitability in Asia (Yonow, Kriticos & Ota, 2017), except for insular Indonesia where it was only introduced at one site in late 2014. The far superior P. manihoti infestation pressure in eastern Indonesia (i.e., NTT, Lombok), where A. lopezi waits to be introduced, further emphasize the contribution of this parasitoid to mealybug control. Across locations, A. lopezi attained maximum parasitism levels of 97% (in late dry season, at Tay Ninh), which greatly surpass the 33–36% established threshold (of maximum parasitism rate) for successful biological control—i.e., which results in lasting pest population suppression (Hawkins & Cornell, 1994). At multiple sites, parasitism rates equally surpassed (max. 30%) levels from smallholder plots in Africa’s savanna (Hammond & Neuenschwander, 1990). Factors ensuring this exceptional parasitoid efficacy and resulting pest control are: (a) unique features of the cassava crop, including prolonged durational stability, vegetational complexity and a constitutive secretion of energy-rich nectar for foraging parasitoids (Pinto-Zevallos, Pareja & Ambrogi, 2016); (b) spatio-temporal continuity of mealybug-infested crops at a landscape level (Schellhorn, Bianchi & Hsu, 2014), especially in locations where farmers employ staggered planting and piece-meal harvesting; (c) favorable ecological traits of A. lopezi, including high dispersal ability, environmental adaptability and density-dependent parasitism (Neuenschwander et al., 1989); (d) non- or limited use of (prophylactic) insecticides, except for Thailand and parts of southern Vietnam; and (e) human-assisted dispersal of A. lopezi, via (parasitized) mealybug-infested planting material (Herren et al., 1987). Furthermore, substantial fertilizer inputs and suitable water management in areas with intensified cassava production—e.g., Vietnam’s Tay Ninh province, or parts of eastern Thailand such as Nakhon Ratchassima- likely benefited biological control further by boosting A. lopezi development and fitness (Wyckhuys et al., 2017a). All of the above factors may have contributed to the stabilization of mealybug populations at 27.2% incidence in affected fields and field-level abundance below 10–15 individuals per infested tip, similar to what was observed in Africa during the 1980s campaign (Fig. 2; Hammond & Neuenschwander, 1990).

Exclusion cage assays illustrate how biological control enabled a root yield recovery of 5.3–10.0 t/ha in two main cassava varieties, and how 2015 yields under ‘no cage’ (i.e., ‘real-world’) conditions were in line with historic in-country yield tendencies. Though no direct field-level measurements were made of A. lopezi parasitism during the cage trials, biological control was found to occupy a central role in lowering P. manihoti populations (Thancharoen et al., 2018), and A. lopezi is a determining factor in ensuring mealybug suppression in other key cassava-growing areas in Southeast Asia (Le et al., 2018). Cage trials also showed large variability in responses between the two cassava clones, likely reflective of differences in plant vigor and an individual clone’s photosynthetic capability (Connor, Cock & Parra, 1981; Cock, 2012). The cassava plant does possess a unique set of features to sustain root production under (a)biotic stress, including the adaptive mobilization of biomass and a highly-effective use of resources (Cock, 2012). Yet, the pronounced production losses—under cage conditions, particularly for R72—can be ascribed to continuous (unrestrained) attack of the active apex, direct damage to stems and high rates of plant death.

As P. manihoti presently occurs at low infestation pressure across mainland Southeast Asia, cage assays lend themselves to further extrapolation to a broader geographical scale. Yet, local environmental conditions may still affect parasitoid abundance, efficacy and (biological control) impact. For example, slightly higher P. manihoti population levels were recorded in settings with sandy, low-fertile soils (Wyckhuys et al., 2017a) and biological control under those conditions merits closer research attention. Also, as landscape composition and plant disease infection status modulate P. manihoti performance and efficacy of biological control at a local scale (Wyckhuys et al., 2017b; Le et al., 2018), further replicated trials could be warranted to validate the robustness of our findings under varying agro-ecological contexts. Despite the above confounding factors, careful analysis of production statistics and commodity market fluxes (as in section iv and Wyckhuys et al., 2018) support our empirical results and confirm A. lopezi to be the major biotic factor affecting mealybug population growth. In the meantime, Indonesian sites where A. lopezi has not yet established now constitute a ‘natural laboratory’ to refine and validate existing projections on A. lopezi-mediated yield gain (and crop recovery), in advance of the natural arrival/introduction of the parasitoid.

In tropical Asia, cassava underpins a multi-billion dollar starch sector, constitutes a key source of farm income and provides an (oftentimes indirect) means to food security for poor, under-privileged populations (Howeler, 2014; Delaquis, De Haan & Wyckhuys, 2017). On the one hand, the P. manihoti-induced yield shocks, as recorded during 2009–2011, can have major implications for rural livelihoods. Sustained pest attack can aggravate food security issues in areas where cassava is a prime food staple or progress into chronic ‘poverty traps’ (Tittonell & Giller, 2013), all of which is counteracted through A. lopezi-mediated biological control. Aside from restoring FRY, A. lopezi equally helped recover a plant’s total dry matter or ‘biological yield’ (Thancharoen et al., 2018), which is particularly relevant in areas of tropical Asia where cassava leaves and shoots constitute part of the local diet of rural communities. On the other hand, the net productivity loss of 5.14 million ton of fresh root equaled a respective loss of revenue of US$ 267.5–591.7 million (at 2009–10 factory price) for Thailand’s cassava sector and the Asia-based starch industry. Hence, aside from its manifest (direct, indirect) contribution to local food security, socio-economic benefits of the P. manihoti campaign are deemed to be substantial and potentially equal or even surpass those recorded previously in Africa (Zeddies et al., 2001).

Yield recovery levels in our cage assays were considerably higher than the 2.5 t/ha yield increase recorded through on-farm measurements in sub-Saharan Africa (Neuenschwander et al., 1989). At Thai farm-gate prices, A. lopezi-mediated yield recovery equals to US$200-704 per ha (Thancharoen et al., 2018), though this does not take into account changes in production costs, local elasticities of supply and demand, or (often substantial) insecticide/fertilizer expenditures. Though we do call for caution in extrapolating our findings, the approximate value of P. manihoti biological control in Asia could thus be hundreds of dollars higher than existing estimates of $63 ha−1 year−1 across global biomes including natural systems (Costanza et al., 1997), $33 ha−1 year−1 for (natural) biological control of the soybean aphid in the US Midwest (Landis et al., 2008), or $75 to $310 ha−1year−1 for bird-mediated pest control in Costa Rican coffee (Karp et al., 2013). From the above it is clear that the potential of insect biological control has been significantly under-valued (Landis et al., 2008; Naranjo, Ellsworth & Frisvold, 2015), and that comprehensive cost-benefit analyses are urgently needed to raise (or restore) societal recognition of this important and safe ecosystem service.

These substantial economic benefits of (naturally-occurring, cost-free) biological control need to be contrasted with the unrelenting global increase in the use of insecticides for mitigation of (domestic and, increasingly invasive) pests (Enserink et al., 2013). Following the P. manihoti invasion, pesticides have equally become pervasive in Thailand’s cassava crop and growers have embraced the (prophylactic) use of neonicotinoid insecticides. Yet, given the omnipresence of A. lopezi and the largely low mealybug population levels in pesticide-free cassava plots across Southeast Asia, cost-effectiveness of such approaches needs closer scrutiny. Though pesticides do bring certain benefits to society, they tend to simplify ecological communities, adversely impact natural enemies and thus accelerate pest proliferation (Lundgren & Fausti, 2015). On the other hand, our work shows how a carefully-selected, specialist natural enemy constitutes a viable alternative to insecticide-centered approaches. Hence, potential (non-target ecological) risks of classical biological control have to be viewed in terms of refraining from action and thus creating room for far-less environmentally-friendly tactics such as (systemic) pesticides (Messing & Wright, 2006; Suckling & Sforza, 2014; Hajek et al., 2016).

Conclusions

This study provides a quantitative assessment of how IBC helped restore primary productivity in Asia’s cassava crop, following the arrival and extensive spread of an invasive sap-feeding pest, i.e., the cassava mealybug. We demonstrate that IBC can provide durable and cost-effective control of the invasive P. manihoti, and delivers substantial and sustainable socio-economic and environmental benefits (Bale, Van Lenteren & Bigler, 2008; Naranjo, Ellsworth & Frisvold, 2015). Furthermore, we emphasize that by recognizing the minimal ecological risks associated with the introduction of a host-specific parasitoid and by deliberately balancing benefits and risks of biological control (see Heimpel & Cock, 2018), the transformative potential of IBC can be fully exploited. In a world typified by massive declines in insect numbers, extreme biodiversity loss, and dwindling public interest in biological control (Bale, Van Lenteren & Bigler, 2008; Hallmann et al., 2017; Warner et al., 2011), our research underlines the immense yet largely untapped potential of ecologically-based approaches to resolve (invasive) pest problems. We advocate that (judiciously-implemented) IBC can positively assist with intensifying global agriculture and help feed a growing world population in the 21st century.

Supplemental Information

Supplemental Information 1 Mealybug distribution records and parasitoid presence

Database with mealybug incidence and abundance records, combined with A. lopezi field-level presence data.

Click here for additional data file.

Supplemental Information 2 Supplementary figures and table

Click here for additional data file.

This manuscript presents original data-sets, generated through fully collaborative research, with trials jointly conceptualized, defined and executed by national program staff and CIAT personnel. This paper is dedicated to the memory of Drs. RJ O’Neil, A. Winotai and AC Bellotti—pioneers in the field of insect biological control. We are grateful to collaborators at China Academy of Tropical Agricultural Sciences CATAS, for sharing monitoring records from Yunnan province.

Additional Information and Declarations

Competing Interests

Author Contributions

Data Availability

Steve Wratten is an Academic Editor for PeerJ, and Jonathan Lundgren is an employee of the Ecdysis Foundation.

Kris A.G. Wyckhuys conceived and designed the experiments, analyzed the data, prepared figures and/or tables, authored or reviewed drafts of the paper, approved the final draft.

Prapit Wongtiem conceived and designed the experiments, authored or reviewed drafts of the paper, approved the final draft.

Aunu Rauf conceived and designed the experiments, contributed reagents/materials/analysis tools, authored or reviewed drafts of the paper, approved the final draft.

Anchana Thancharoen conceived and designed the experiments, performed the experiments, analyzed the data, contributed reagents/materials/analysis tools, prepared figures and/or tables, authored or reviewed drafts of the paper, approved the final draft.

George E. Heimpel, Geoff M. Gurr, Jonathan G. Lundgren, Vi X. Le, Matthew J.W. Cock, Teja Tscharntke, Steve D. Wratten, Minsheng You, Yanhui Lu, Johannes W. Ketelaar, Georg Goergen and Peter Neuenschwander authored or reviewed drafts of the paper, approved the final draft.

Nhung T.T. Le conceived and designed the experiments, performed the experiments, contributed reagents/materials/analysis tools, authored or reviewed drafts of the paper, approved the final draft.

Muhammad Zainal Fanani performed the experiments, analyzed the data, authored or reviewed drafts of the paper, approved the final draft.

Dharani D. Burra analyzed the data, authored or reviewed drafts of the paper, approved the final draft.

Leo K. Palao and Glenn Hyman analyzed the data, prepared figures and/or tables, authored or reviewed drafts of the paper, approved the final draft.

Ignazio Graziosi performed the experiments, analyzed the data, authored or reviewed drafts of the paper, approved the final draft.

Liem V. Nguyen contributed reagents/materials/analysis tools, authored or reviewed drafts of the paper, approved the final draft.

The following information was supplied regarding data availability:

The raw data are provided in a Supplementary File.

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
