# Peer review of "Continental-scale suppression of an invasive pest by a host-specific parasitoid underlines both environmental and economic benefits of arthropod biological control"

_PeerJ, doi:10.7717/peerj.5796_

## Round 0.1 · original submission · Major Revisions

Your manuscript has been seen by 3 reviewers and myself. All acknowledge on the quality and relevance of your work. I agree. Reviewers 1 and 3 raise important methodological issues that should be sorted out before proceeding any further, in order to guarantee reproducibility and clarity and robustness of your results.

·

Basic reporting

Please see "General comments for the author"

Experimental design

Please see "General comments for the author"

Validity of the findings

Please see "General comments for the author"

Additional comments

This study presents an impressive amount of data on an important invasive pest and its control using a classic biological control approach (i.e. introduction of the natural enemy from its origynal area of distribution). The authors have followed the dynamics of the insect pest and the parasitoid since the introduction of the natural enemy. These data are followed by experiments in which natural enemies are excluded to assess their impact in controlling the pest. Finally, the authors perform an economic model to assess the economic benefit associated to the introduction of the natural enemy. The paper is mostly well written and the topic suits the journal. Reports of these kind of "natural experiments" are less common than they ought to, and can provide important insights into the ecology of insects and their natural enemies and their applied benefits. Despite these merits I recommend a major revision as I have identified a series of serious concerns that I detail below. Given these concerns at the level of the experimental design, analyses and their interpretation I won't comment on the discussion section.

1. My main concern is about the parasitoid exclusion experiment (L235-236). If I understood it well, this experiment had four replicated plants per treatment (exclusion with cage, non exclusion with cage, non-exclusion without cage), which were enclosed in the same cage. If this is the case, I believe data is clearly pseudoreplicated (even if cage is included as random effect in the statistical model) as plants within the same cage cannot be considered as independent.

2. More details on the analyses performed are needed. In L203- L214, why was field included as random in one model and as a fixed effect in the other? In L203 sampling date was included as fixed factor, but it is not clear to me why. Usually fixed factors are included in models to test specific hypothesis. Was this the case, or was this variables included to correct for the non-independence of samples taken the same day? In L114, how are these models built, and which was the unit of replication?

I am not sure I understand the model described in L266-273: data on productivity per region was correlated with different environmental variables, and year was also included as explanatory variable. Then in 2011 and 2012 the categorical variable "parasitoid" was included as being present while from 2008 to 2010 the parasitoid was included as absent. I can see the point behind this analysis, but I find a bit unrealistic that the authors consider that the parasitoid arrived exactly at the same time in all regions. Do the authors have more finely detailed data?

3. Many correlations between host density (or host population growth) and parasitism are performed to infer parasitoid control over its host (L175, L208-210, L313, ...). To my view these types of correlations do not represent parasitoid control over the host but whether the parasitoid responds to its host in a density-dependent manner.

Some minor comments follow:
1. In L218 the authors state "In August 2014, a field study was initiated at the Rayong Field Crops Research Center in Rayong, Thailand (Thancharoen et al., under review)". Does this imply that the results presented here are submitted elsewhere? Please clarify.

2. The title and the abstract should be toned down, I find that the sentence "... heralds a new era for
arthropod biological control" in the title overstates the results of this study. The same applies to the abstract.

3. Although I am not a native English speaker, some sentences could be written in a more formal manner. Examples include L105 "One widely-acclaimed IBC program is the Africa-wide initiative targeting the invasive cassava mealybug", or L140 "We present a data-rich body".

4. L326 and Figure 2. I don't see the point in presenting here data from the population densities in Nigeria in 1982. This data should be used in the discussion section.

Reviewer 2 ·

Basic reporting

This research documents and quantifies the enormous beneficial impact an importation biological control program has had on suppressing populations of the invasive and highly destructive cassava mealybug in southeast Asia.

The work is very well done, the results are compelling, and the graphics clearly show the impact the introduced parasitoid A. lopezi had on suppressing pest populations and the concomitant increases in cassava yields.

Importantly, in addition to population impact monitoring/assessments, the authors have provided solid and credible economic estimates supporting the financial benefits of this program to agriculturists, the majority of whom are small landholders that suffer significant income reductions because of invasive pests.


I've made numerous editorial suggestions/edits/comments on the paper - including the title, which in my opinion is a little too grandiose!

Experimental design

Very well done, I don't have any significant criticisms. The authors have employed well accepted experimental designs and statistical analyses. These all appear appropriate to me.

Validity of the findings

The authors have reached well reasoned conclusions based on the experimental data that has been collected and analyzed. A major strength of this work is the vast distances over which it was conducted and the number of years over which data were collected. When combined with manipulative field experiments (i.e., the use of cages) and the estimated economic benefits these findings are strong and credible.

Additional comments

I have no additional comments to share. The work is very well done, the team is highly respected and renowned for the work they do on invasive pest management. This is an excellent piece of work this is a "classic" textbook example of the ability of upper trophic level organisms to suppress invasive pest population thereby supporting "enemy release" theory.

Annotated reviews are not available for download in order to protect the identity of reviewers who chose to remain anonymous.

Reviewer 3 ·

Basic reporting

The article presents the results of field surveys and experiments well designed and executed. The introduction and discussion are clear and well written, with abundant and relevant references. The raw data is provided and the figures are generally clear (but see comments below). The main issue with the paper is the inappropriate/incomplete description of the statistical methods used and the results obtained, which prevents judging the validity of the conclusions of the study (see below details).

Experimental design

The experimental design is adequate, but there are several major issues on the way analysis are described and results are reported that need to be fixed before the article can be published, to properly judge the validity of the results presented. I list some below:
Figure 2 is confusing. Why are less columns in b than a? Are a and b based on different samplings, fields? Pleas clarify. Also, considered using the bars for the same variable, i.e. mealybug abundance in Vietnam in both a and b (currently in a is for Nigeria and in b is for Vietnam). Indicate in the graph (or graph caption) when A. lopezi was released in Nigeria and detected in Vietnam.
L332 – 334 and Figure 3. The statistical analysis on this data is very unclear. You state that field had not significant effect, but you corrected per field, why? Also, the figure suggests you fit models with different slopes per field, why? How do these models relate to the statistic presented in the text? Why did you report a chi square test? Please explain the statistic used to assess the correlation between mealybug growth rates and parasitism and how you arrive to 87 degrees of freedom. Please provide a more clear and detailed information in the methods section and describe appropriately in results. Remove significantly in Line 332.
L342 – 351 and Figure 4. Please clarify if the CMD in the figure reports final values or average across sampling dates. What is the rationale to analyzed CMD over time? Typically, I expect an analysis of mean abundance in each date interacting with time, but only the final date analyzed for a cumulative density (i.e. a cumulative density is constructed in a way that will interact with time and will involve analyzing the same counts multiple times…). Please indicate clearly the rationale for your analysis, clarify that the variable analyzed meet the assumptions of the statistical model and provide references of previous studies supporting this analytical approach. The large number of degrees of freedom (189) suggest pseudo-replication, how was the field effect accounted for? In traditional repeated measures models, the denominator df for the time terms is lower than for the main effects, please explain how you have the same df for time and treatment effects in your models. You indicate that full cage differ from sham and no cage, but there are no tests comparing means reported, please provide statistical support to these differences.
There are 3 df of freedom for treatments, yet there are 3 treatments described and presented in the results (full cage, sham cage and no cage. Only in L 352 you mentioned 4 treatments, but I can’t find any other reference to an additional treatment. Please clarify this crucial point.
L352-359. You provide a Kruskal-Wallis test, include description in the methods on the rationale of using this method. There are no mean tests to separate the 3 treatments; these tests need to be provided to compare treatments after ANOVA or KW overall significance tests.
L264. Please explain how weighted means were calculated.
L371-379. Multiple regression models don’t allow categorical variables (such as Presence / absence of the parasitoid), this must be a GLM; please clarify… The 3 df in L377 are difficult to understand, where 4 periods of time analyzed in that period?? The analysis is very unclear here, please provide more details and complete statistics for the models used in a supporting information.

Validity of the findings

The main claim in the in paper is that the introduction of A. lopezi resulted in a continental-scale reduction in P. manihoti damage and yield losses. The only empirical evidence supporting this claim is the negative correlation between mealybug growth rates and parasitism level, but this evidence is restricted to fields in Vietnam. The manipulative study with cages does not report parasitism levels, so precludes the authors to claim that parasitism caused the reduction on CMD. The third piece of information available is the multiple regression analysis that incorportate multiple factors to explain cassava yield, including presence/absence of A. lopezi. However, this analysis is poorly presented and it is difficult to judge if analyses were appropriate and to assess the importance of each factor tested. I suggest that the authors should improve the description of the models and their results presented in section iv to fully support the claims in the paper.

Additional comments

Minor comments:
Citations in the results section are usually inappropriate and belong to the discussion section of the paper (i.e. L 365)
L370. What is US$ FOB?

---

## Round 0.2 · Minor Revisions

This is a much improved version of the previous manuscript. The manuscript has been seen by one of our previous reviewers and myself. Modifications at this stage are minor. Please follow the reviewer's suggestions.

·

Basic reporting

see general comments

Experimental design

see general comments

Validity of the findings

see general comments

Additional comments

This is the second time I read this manuscript, and I appreciate the effort made by the authors to improve it. Most of my concerns have been addressed, but I recommend a minor revision based on the comments below.

Abstract, L 45 and 49 are a bit overstating what is tackled in this review. Please tone down these sentences a bit. I don't think biocontrol is the prime ecosystem service on Earth (think of for example oxygen provision by plants and algae). I also don't agree biocontrol needs a reform, it needs more interesting cases like the one presented here.

L81. "In tropical terrestrial ecosystems, these trends might be even more pronounced though they routinely remain undocumented". Can the authors explain why?

L234-238. Based on the main text, the unit of replication in the exclusion experiment is not yet clear, although in the rebuttal letter the authors clearly show that pseudoreplication is not an issue. Can the authors be more clear in the main text and specify that each treatment had four replicates, each consisting of one cage containing four plants?

L324-326. "In fields where A. lopezi had effectively established, mealybug pest pressure was significantly lower at increasing levels of parasitism". I'd say that mealybug density correlated with parasitism, the speculation that this is linked to lower pest suppression should be mentioned in the discussion section.

L337-339. Move these types of data interpretation to the discussion section.

---

## Round 0.3 · accepted · Accept

The authors have addressed satisfactorily the minor comments raised by the referee. I am happy to accept this very exciting paper.

#